# TLR Responses in Preterm and Term Infant Cord Blood Mononuclear Cells

**DOI:** 10.3390/pathogens12040596

**Published:** 2023-04-14

**Authors:** Jeremy Anderson, Georgia Bender, Cao Minh Thang, Le Quang Thanh, Vo Thi Trang Dai, Phan Van Thanh, Bui Thi Hong Nhu, Do Ngoc Xuan Trang, Phan Thi Phuong Trinh, Nguyen Vu Thuong, Nguyen Trong Toan, Kim Mulholland, Daniel G. Pellicci, Lien Anh Ha Do, Paul V. Licciardi

**Affiliations:** 1Murdoch Children’s Research Institute, Melbourne, VIC 3052, Australia; 2Department of Paediatrics, University of Melbourne, Melbourne, VIC 3010, Australia; 3Pasteur Institute of Ho Chi Minh City, Ho Chi Minh City 72408, Vietnam; 4Tu Du Hospital, Ho Chi Minh City 700000, Vietnam; 5Department of Infectious Disease Epidemiology, London School of Hygiene and Tropical Medicine, London WC1E 7HT, UK; 6Department of Microbiology and Immunology, University of Melbourne, Melbourne, VIC 3010, Australia

**Keywords:** infant, preterm, TLR, inflammation, immune response

## Abstract

Preterm infants are more susceptible to severe bacterial and viral infectious diseases than their full-term counterparts. A major contributor to this increased susceptibility may be due to differences in their ability to respond to pathogens. While studies have demonstrated altered bacterial Toll-like receptor (TLR) responses, there is limited data on viral TLR responses in preterm infants. In this study, cord blood mononuclear cells (CBMCs) from 10 moderately preterm (30.4–34.1 wGA), 10 term (37–39.5 wGA) infants, and 5 adults were stimulated with TLR2 (lipoteichoic acid), TLR3 (poly I:C), TLR4 (lipopolysaccharide), TLR7/8 (R848), and TLR9 (CpG-ODN 2216) agonists. Following stimulation, the cellular response was measured by intracellular flow cytometry to detect cell-specific NF-κB (as a marker of the inflammatory response), and multiplex assays were used to measure the cytokine response. This study found that preterm and term infants exhibit very similar baseline TLR expression. In response to both bacterial and viral TLR agonists comparing cell-specific NF-κB activation, preterm infants exhibited increased monocyte activation following LTA stimulation; however, no other differences were observed. Similarly, no difference in cytokine response was observed following stimulation with TLRs. However, a stronger correlation between NF-κB activation and cytokine responses was observed in term infants following poly I:C and R848 stimulation compared to preterm infants. In contrast, despite similar TLR expression, adults produced higher levels of IFN-α following R848 stimulation compared to preterm and term infants. These findings suggest preterm and term infants have a similar capacity to respond to both bacterial and viral TLR agonists. As preterm infants are more likely to develop severe infections, further research is required to determine the immunological factors that may be driving this and develop better interventions for this highly vulnerable group.

## 1. Introduction

Preterm infants are those born before 37 weeks of gestation and account for 15 million births each year globally [1]. Preterm infants are often categorised as extreme (<30 weeks), moderate (30–34 weeks), or late (35–36 weeks), with approximately 84% of preterm births in the moderate-to-late stage [2]. One of the main issues with prematurity is the increased risk of bacterial and viral infections [3]. Preterm infants have an increased susceptibility to severe bacterial and viral diseases. These include sepsis [4], necrotizing enterocolitis [5], and severe lower respiratory tract infections [6]. While susceptibility to severe disease is most common in extreme preterm infants, moderate preterm infants also suffer from severe disease and contribute significantly to the disease burden in this population [7,8].

A reason for their increased susceptibility to severe disease outcomes may lie in their relatively immature innate immune system [9]. Studies have suggested that preterm infants have a reduced capacity to recognize and respond to invading pathogens, which may enhance their susceptibility to severe infections [10]. This has been demonstrated in studies comparing the Toll-like receptor (TLR) response in preterm and term infants. TLRs are responsible for detecting bacterial (TLRs 1, 2, 4, 5, and 6) or viral (TLRs 3, 7, 8, and 9) ligands [11]. Upon recognition, this leads to the activation of nuclear factor-κB (NF-κB) to initiate inflammatory responses against pathogens, which can be protective or harmful [12]. Studies have shown conflicting results following TLR stimulation. These have primarily focused on bacterial ligands such as TLR2 (lipoteichoic acid) or TLR4 (lipopolysaccharide) and suggest preterm infants either have a reduced capacity to produce inflammatory cytokines or exhibit no differences compared with full-term infants [13,14,15,16]. There are limited studies for viral TLR ligands, suggesting preterm infants exhibit a weaker inflammatory response compared to term infants [14,17,18]. However, further research in this area is required.

This study aimed to investigate cell-specific NF-κB activation and cytokine responses following bacterial and viral TLR stimulation of preterm and term cord blood mononuclear cells (CBMCs) to provide greater insights into potential differences in their response to pathogen stimulation.

## 2. Materials and Methods

### 2.1. Study Design and Participants

Cord blood was obtained from a cohort of 10 healthy preterm (30.4–34.1 wGA) and 10 term (37–39.5 wGA) infants from Tu Du Hospital, Ho Chi Minh City, Vietnam [19]. All infants in this study were born by normal vaginal delivery and were free of early-onset neonatal infection. Cord blood was not collected from infants who had:Major or suspected major malformations, including congenital heart disease, genetic syndromes,Clinical evidence of chorioamnionitis,Rupture of membranes for more than 24 h,Suspected or confirmed early-onset sepsis.
Or if their mothers had:Autoimmune disease, immunodeficiency syndrome, immunosuppressant, or immunomodifying treatment for more than 3 months.Infection: human immunodeficiency virus, hepatitis B, hepatitis C, or primary herpes simplex virus infection during current pregnancy.Physical, psychiatric, or complex social situations where the mother and baby may not be able to fully participate, such as maternal alcohol or substance dependency, or issues about child protection.

The cord blood samples were transported to the Pasteur Institute of Ho Chi Minh City, Vietnam, for cord blood mononuclear cells (CBMCs) isolation and serum separation within 4 h of collection. Briefly, CBMCs were isolated by density gradient centrifugation using Lymphoprep (Axis-Shield, Oslo, Norway) at 400× *g* for 30 min without brake. Isolated CBMCs were washed twice with PBS prior to being resuspended in freezing mix (90% foetal bovine serum (FBS) + 10% DMSO) and stored at −80 °C for 24 h in a cool cell (Sigma-Aldrich, St. Louis, MO, USA). This was then transferred to liquid nitrogen. Sera was stored at −80 °C. Samples were then shipped to the Murdoch Children’s Research Institute in Melbourne, Australia. This study was approved by the Pasteur Institute Ho Chi Minh City Ethics Committee (Ethics approval: 213/QD-PAS) and the Royal Children’s Hospital (RCH) Human Research Ethics Committee (HREC; 56904).

In addition, peripheral blood from five healthy adults was collected at the MCRI following informed consent and was approved by the Royal Children’s Hospital (RCH) Human Research Ethics Committee, Melbourne, Australia (HREC; 35253). Peripheral blood mononuclear cells (PBMCs) were isolated following a similar protocol to our CBMC isolation immediately after collection and stored in liquid nitrogen.

### 2.2. Study Reagents

RPMI-160, FBS, L-glutamine, penicillin-streptomycin, and R_10_ media (RPMI-1640 medium supplemented with 10% FBS, 2 mM L-glutamine, and 1000 IU penicillin-streptomycin) were purchased from Sigma-Aldrich, St. Louis, MO, USA. Anti-mouse compensation beads were purchased from BD Bioscience (San Diego, CA, USA). All flow cytometry antibodies used and their suppliers are indicated in Appendix A.

### 2.3. Stimulation Assay

PBMCs and CBMCs were thawed at 37 °C in 10 mL of R_10_ media and centrifuged at 500× *g*, 22 °C for 5 min. Supernatants were then removed, and cells were suspended in 1 mL of R_10_ and counted using a TC20 automated cell counter with trypan blue. Cells were then diluted to achieve 1 × 10^6^ cells in 900 µL of R_10_ media.

1 × 10^6^ PBMCs and CBMCs were left unstimulated (media only) or stimulated with 100 µL TLR ligands (Table 1). For intracellular NF-κB detection, bacterial TLR ligands (LTA (TLR2), LPS (TLR4)) were used to stimulate cells for 2 h and viral TLR ligands (Poly I:C (TLR3), R848 (TLR7, TLR8), and CpG-A ODN 2216 (TLR9)) for 4 h at 37 °C, 5% CO_2_ [20,21]. For cytokine measurement, cells were incubated for 24 h at 37 °C with 5% CO_2_ following stimulation.

### 2.4. Flow Cytometry

Following incubation, cells were centrifuged at 400× *g* at 4 °C for 4 min and then washed with 1 mL of flow buffer containing PBS supplemented with 2% FBS and 2 mmM of Ethylenediaminetetraacetic acid (EDTA). Each sample was stained with 50 µL of surface antibody cocktail 1 (containing TLRs 2 and 4) for 20 min on ice, then cells were resuspended in 1 mL flow buffer and centrifuged at 400× *g*, 4 °C for 4 min (Appendix A). Cells were then fixed with 100 µL of fixation buffer for 20 min on ice before being washed twice with 1 mL of permeabilisation buffer (diluted 1:10 with distilled water). After that, cells were stained with 50 µL of intracellular antibody cocktail 2 (containing TLRs 3, 7, and 9) and incubated on ice for 20 min (Appendix A). Cells were then washed with 1 mL of permeabilisation buffer (1:10 diluted in distilled water; BD Bioscience, San Diego, CA, USA), followed by 1 mL of flow buffer, and centrifuged between washes at 400× *g*, 4 °C, for 4 min. Cells were then resuspended in 100 µL of flow buffer for acquisition on the Cytek Aurora (Cytek Biosciences, CA, USA). Compensation was performed prior to acquisition using compensation beads. Analysis was performed using Flowjo v10.7.1 software (FlowJo, LLC), and gating strategies are shown in Appendix A.

### 2.5. Cytokine Measurement

A commercial multiplex bead array kit (8-plex human cytokine assay; Bio-Rad, NS, Australia) was used to measure IL-2, IL-4, IL-6, IL-8, IL-10, TNF-α, IFN-γ, and GM-CSF from supernatants according to the manufacturer’s instructions. Results were analysed on a Luminex 200 instrument (Luminex, Austin, TX, USA) fitted with the Bio-Plex Manager Version 6 software, and results were reported in pg/mL.

For IFN-α measurement, a commercial enzyme-linked immunosorbent assay (ELISA) kit (R&D systems, Minneapolis, MN, USA) was used according to the manufacturer’s instructions. The plate was read immediately after the assay at 450 nm (reference wavelength of 630 nm) using a microplate reader (BioTek, Winooski, VT, USA) and values were reported in pg/mL.

### 2.6. Statistical Analysis

To compare preterm infant, term infant, and adult cell-specific NF-κB expression (median fluorescence intensity (MFI)) and cytokine production, a Kruskal-Wallis test with a Dunn’s post-hoc test was used, and data were presented as boxplots showing the median ± interquartile range (IQR) with minimum and maximum whiskers. Cellular NF-κB expression (MFI) was presented on boxplots as background-corrected values, calculated by subtracting NF-κB MFI in a TLR-stimulated sample from NF-κB MFI in its respective unstimulated sample. Cytokine responses were presented using boxplots calculated as fold change values. Correlation matrices between cytokine response and NF-κB expression were created using RStudio v4.1.22 (Boston, MA, USA). All other analyses were performed using GraphPad v8.1 (Boston, MA, USA).

## 3. Results

### 3.1. Preterm and Term Infant Characteristics

In this study, 10 healthy preterm (30.4–34.1 wGA) and 10 healthy term (37–39.5 wGA) infants were studied. All cord blood was collected from infants delivered vaginally, and there was no clinical chorioamnionitis detected (Table 2). The age range of adults that participated in this study was 28–40 years, with two males and three females.

### 3.2. TLR Expression in Preterm and Term Infants

To observe differences in baseline (unstimulated) TLR expression between preterm and term infants, cell-specific TLR expression was determined by flow cytometry. Although we tested a range of TLRs (TLR2, TLR3, TLR4, TLR7, and TLR9), TLR7 staining was unreliable, therefore we did not include it in the analyses. No differences were observed in the frequency of TLR2, TLR3, TLR4, and TLR9 on monocytes between preterm infants, term infants, and adults (Figure 1A). Similarly, no differences were observed between preterm infants, term infants, and adults for TLR2, TLR3, and TLR4 expression on myeloid dendritic cells (mDCs) and TLR9 expression on plasmacytoid dendritic cells (pDCs) (Figure 1B,C). While no differences were observed in natural killer (NK) cell TLR2 and TLR3 between all groups, we found a higher expression of TLR9 on NK cells in adults compared to preterm infants (*p* < 0.05; Figure 1D).

### 3.3. TLR-Mediated NF-κB Activation in Preterm and Term Infants

To observe differences in TLR-mediated immune signalling between preterm and term infants, cell-specific intracellular NF-κB activation was assessed following TLR stimulation by flow cytometry. Following stimulation with LPS, preterm infants had a significantly higher NF-κB MFI compared to adults (*p* < 0.01) but not term infants in mDCs. No differences were observed between preterm infants, term infants, and adults for all other TLRs in mDCs. A higher NF-κB MFI in term infants compared to adults was observed in pDCs following R848 stimulation (*p* < 0.05); however, no other differences were observed in this population. In monocytes, adults expressed a higher NF-κB MFI compared to term infants following LPS and LTA stimulation (*p* < 0.05), whereas preterm infants expressed a higher NF-κB MFI compared to term infants in monocytes following LTA stimulation (*p* < 0.05). No differences were observed in monocytes for other TLR stimulation conditions. In natural killer (NK) cells, adults expressed a higher NF-κB MFI following LPS stimulation compared to preterm (*p* < 0.01) and term infants (*p* < 0.05), and a higher NF-κB MFI following LTA stimulation compared to term infants (*p* < 0.05). No other differences were observed for the viral TLRs (Figure 2).

### 3.4. TLR-Mediated Cytokine Responses in Preterm and Term Infants

We next measured the cytokine levels in supernatants from mononuclear cells stimulated with TLR ligands for 24 h. In response to LTA stimulation, preterm infants exhibited a higher fold change for IL-4 and IL-8 compared to adults (*p* < 0.05), but not term infants. Following stimulation with R848, preterm infants exhibited a higher fold change for IL-4, IL-6, and IL-8 compared to adults (*p* < 0.05). Following stimulation with R848 both preterm and term infants had a significantly lower fold change for IFN-α compared to adults (*p* < 0.05). No other differences were observed for IL-2, IL-4, IL-6, IL-8, IL-10, GM-CSF, TNF-α, IFN-γ, and IFN-α following TLR stimulation between preterm infants, term infants, and adults (Figure 3).

### 3.5. Correlating TLR-Mediated NF-κB Activation and Cytokine Responses in Preterm and Term Infants

To investigate the level of coordination seen in preterm and term infant TLR responses to bacterial and viral ligands, we performed unsupervised correlation analyses (Figure 4). Overall, similar patterns of coordination between cellular NF-κB activation and cytokine responses to LPS, LTA, and CpG stimulation were observed between preterm and term infants. However, following poly I:C stimulation, there was a larger correlation between cytokine responses in term infants compared to preterm infants, and following R848 stimulation, a stronger correlation between cellular NF-κB and IL-2, IL-4, and IL-6 was observed in term infants compared to preterm infants. In all stimulation conditions, adults exhibited a different response compared to both preterm and term infants.

## 4. Discussion

Severe infection is one of the leading causes of death in children under 5 years of age, especially in low- and middle-income countries (LMICs) [22]. Preterm infants are among the most susceptible to severe infection; however, the factors that confer this vulnerability are poorly understood. As most severe infections occur during early life, studies have endeavoured to understand how preterm innate immune defence mechanisms contribute to infectious disease susceptibility. Importantly, the TLR response is a key area of interest as differences in TLR recognition are observed between neonates and adults [23]. TLRs play a critical role in innate immunity by detecting potentially harmful invading micro-organisms and initiating immune responses where appropriate [20]. This study found that preterm infants exhibit similar responsiveness to both bacterial (LPS, LTA) and viral (poly I:C, R848, CpG) TLR agonists, although the immune coordination among preterm infants was lowest for viral stimuli poly I:C and R848.

Many studies have compared preterm and term infant baseline TLR expression; however, their analysis has been mainly limited to TLR2 and TLR4 expression on monocytes. These studies often report reduced TLR2 and TLR4 expression on monocytes from preterm infants compared to term infants [15,16,24,25,26,27], but some studies found no differences [24,28,29]. In this study, we expanded the repertoire of TLRs and immune cell populations for our analysis to provide a more comprehensive overview of TLR responses in preterm and term infants. However, similar to previous studies, we found no significant differences in bacterial and viral TLR expression on cord blood monocytes, mDCs, pDCs, and NK cells. While TLR9 expression in NK cells was higher in adults compared to preterm infants, this was not associated with any increased IFN-γ following CpG stimulation. This indicates that preterm and term infants may not differ in their ability to recognise bacterial and viral pathogens.

The ability to produce an effective immune response following pathogen recognition is a vital component of immune functionality [20]. NF-κB is an important transcription factor that regulates the expression of a number of genes encoding inflammatory mediators [30]. In this study, we measured p65 to determine NF-κB activation, as it is one of the primary DNA-binding subunits released upon TLR activation [31]. The current study revealed strong NF-κB activation in stimulated monocytes and DCs from preterm infants, term infants, and adults. However, the only difference observed between preterm and term infants was a higher activation of NF-κB in monocytes following LTA stimulation. This is in contrast with the study by Wisgrill et al., in which no difference in monocyte NF-κB expression following LPS and LTA stimulation was seen between preterm (24–32 wGA) and term infants, although the gestational ages in this study were more extreme than ours [24]. However, our findings revealed a trend towards higher NF-κB expression in bacterially stimulated innate immune cells, especially in monocytes from preterm infants. Unlike our study, Nupponen et al. detected a higher NF-κB activation in LPS-stimulated monocytes from the cord blood of preterm infants (27–36 wGA) compared to those from term infants [32]. These findings suggest that, relative to term infants, preterm infants may elicit an over-reactive pro-inflammatory response to bacterial TLRs, which has been implicated in the damaging effects of inflammatory diseases in premature infants [33].

Our study also found that cytokine responses were mostly similar between preterm and term infants in response to bacterial and viral TLRs, consistent with previous studies [14,15]. This is despite the reduced frequencies in mDCs and classical monocytes that we have previously reported [19]. However, other studies have shown decreased cytokine responses in preterm compared to term infants following bacterial and viral TLR stimulation [13,17,25]. A key reason for this may be that in these studies, gestational ages were mostly <30 weeks, compared to our study using 30–34 weeks. This highlights the highly variable findings that exist when comparing preterm and term immune responses to TLRs. Despite this, preterm infants exhibit higher levels of IL-4 and IL-8 in response to LTA and IL-4, IL-6, and IL-8 in response to R848 when compared to adults, whereas term infants do not.

A coordinated inflammatory response is required to effectively prevent infection, and an inability to do this may contribute to the development of severe disease [20]. Following TLR-induced NF-κB activation, inflammatory mediators including cytokines, chemokines, and growth factors are produced [30]. In this study, preterm and term infant TLR-mediated inflammatory responses were mostly similar following LPS, LTA, and CpG stimulation. This is in accordance with other studies observing no difference in cytokine and chemokine responses between preterm and term infants [14,15,17,18,24,28,34,35]. Despite no difference in overall cytokine response, correlating our data to cellular NF-κB activation suggested that the TLR response to poly I:C and R848 was less coordinated in preterm infants compared to term infants. This may be important in the context of viral infection, as poly I:C and R848 are dsRNA and ssRNA agonists, respectively. In dysregulated responses, cytokine production can promote pathology as evidenced through SARS-CoV-2 disease progression, often leading to significant tissue and organ damage brought on by ‘cytokine storm’ [36]. Taken together, these findings suggest that in response to bacterial and viral TLR stimulation, preterm infants show a similar capacity for activation and cytokine production. However, they exhibited reduced coordination following poly I:C and R848 stimulation compared to term infants, which may enhance their susceptibility to severe disease.

Studies have highlighted that the preterm infant gut microbiome is distinct from that of full-term infants, as it contains elevated levels of pro-inflammatory bacteria and is often in dysbiosis [37]. This dysbiosis has been linked to NEC and sepsis. It has been shown that probiotic supplementation with *Lactobacillus* and *Bifidobacterium* species leads to the reduction of NEC in preterm infants and reduced days spent in the hospital [38]. This is suggested to occur due to the suppression of inflammation through NF-κB and the up-regulation of anti-inflammatory genes to improve immune regulation [38]. Therefore, probiotic supplementation in preterm infants may be valuable for improving immune regulation and reducing immunopathology.

A strength of this study is the combination of TLR expression, cell-specific activation through NF-κB and cytokine responses. This generates a more complete picture of potential differences between preterm and term infants in response to bacterial and viral TLRs. Another strength is the similarity between birth mode and the lack of early-onset infection and clinical chorioamnionitis in both the preterm and term groups. This is because these factors are known to influence the immune response [39,40]. The main limitation of this study is the small sample size, which may have reduced the statistical power of our comparative analyses and limited our ability to detect differences between the groups. Another limitation is the use of ex vivo cultures to investigate immunological differences; therefore, the findings may not accurately reflect those observed in vivo. Further studies using purified immune cell populations may be valuable to determine cell-specific responses more accurately. Additionally, examining the relationship between the foetal microbiome and early TLR immune responses would also be worthwhile.

## 5. Conclusions

This study investigated moderate preterm infant and term infant cord blood TLR responses to bacterial and viral ligands. Our data suggests that TLR expression, TLR-mediated NF-κB activation, and cytokine responses were mostly similar between preterm and term infants, although the response to viral ligands in preterm infants was less coordinated. This may partly explain why preterm infants are more susceptible to severe infectious diseases. Further research is needed, particularly in the development of novel interventions, to reduce the burden of severe infectious diseases in this vulnerable group.

## Figures and Tables

**Figure 1 pathogens-12-00596-f001:**
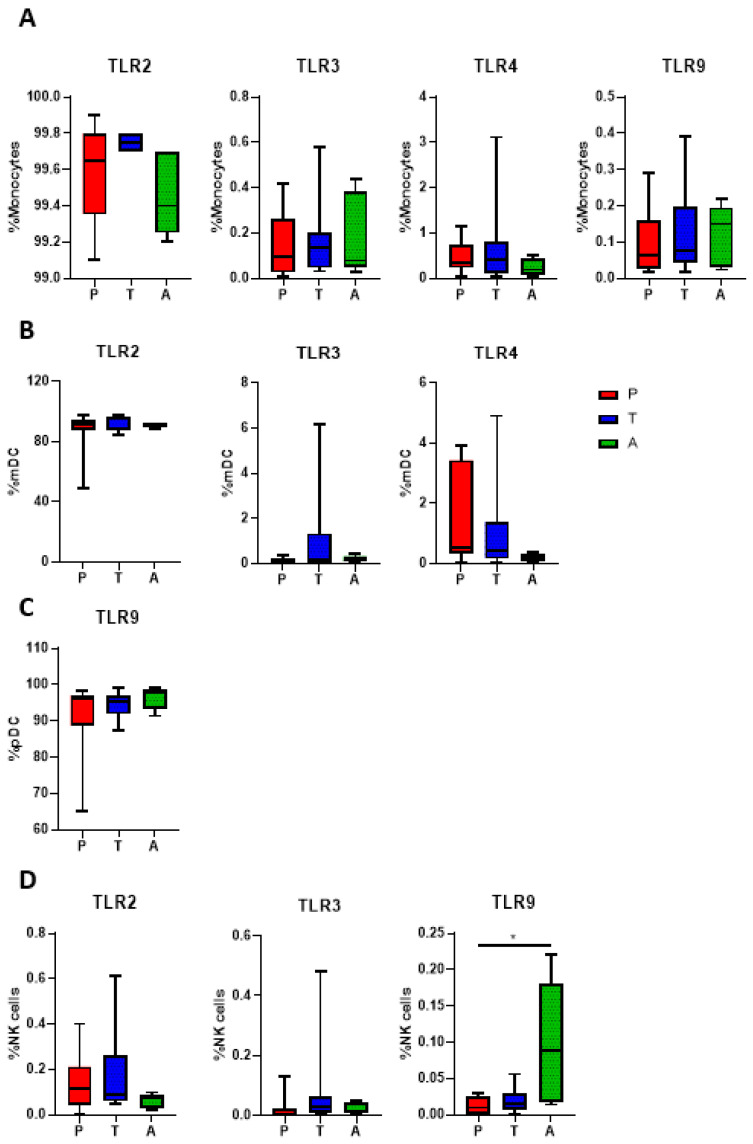
Baseline TLR expression in preterm infants, term infants, and adults. TLR expression in immune cell subsets from preterm infants (P, n = 10), term infants (T, n = 10), and adults (A, n = 5) were determined using flow cytometry. (**A**) TLR expression on monocytes; (**B**) TLR expression on mDCs; (**C**) TLR expression on pDCs; (**D**) TLR expression on NK cells. Data are presented as median ± IQR. Analysis was performed using a Kruskal-Wallis test with a Dunn’s post-hoc test. ** p* < 0.05.

**Figure 2 pathogens-12-00596-f002:**
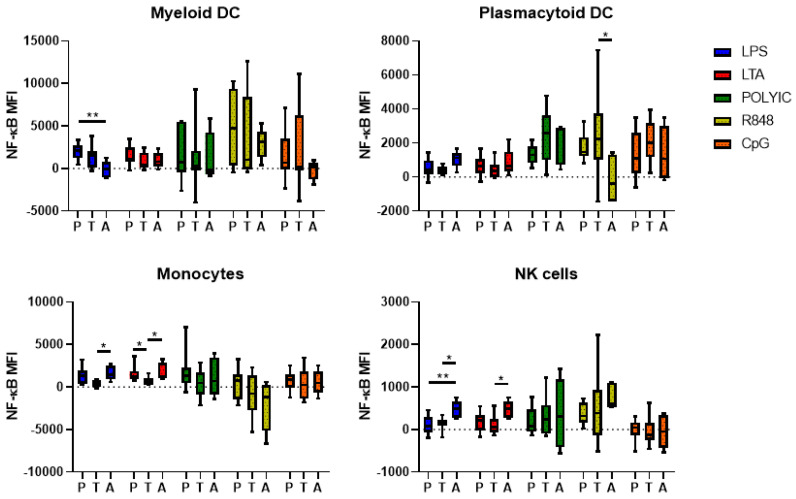
NF-κB activation in preterm infants, term infants, and adults following bacterial and viral TLR stimulation. NF-κB MFI in immune cell subsets from preterm infants (P, n = 10), term infants (T, n = 10), and adults (A, n = 5) were observed using flow cytometry following 2 h stimulation with LPS and LTA and 4 h stimulation with poly I:C, R848 and CpG-ODN 2216. NF-κB MFI values following subtraction of the unstimulated MFI values are presented as the median ± IQR. Analysis was performed using a Kruskal-Wallis test with a Dunn’s post-hoc test. * *p* < 0.05; ** *p* < 0.01.

**Figure 3 pathogens-12-00596-f003:**
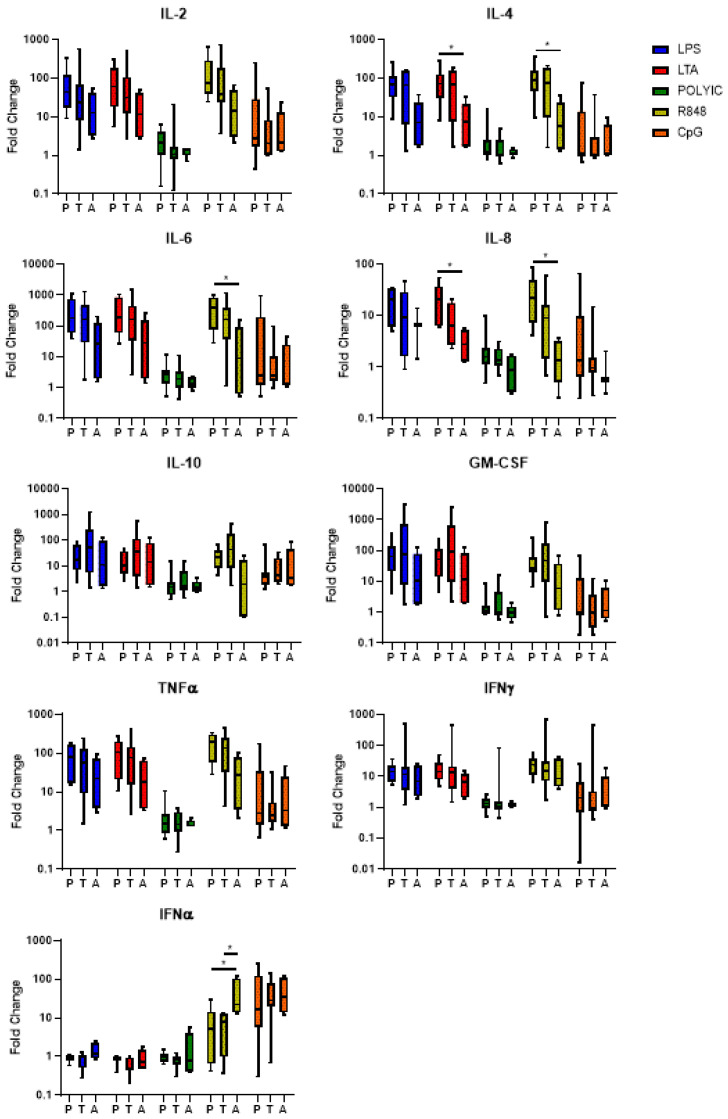
Cytokine levels in preterm infants, term infants, and adults following bacterial and viral TLR stimulation. Following 24 h stimulation with LPS, LTA, poly I:C, R848 and CpG-ODN 2216, supernatant cytokine concentrations from preterm infants (P, n = 10), term infants (T, n = 10), and adults (A, n = 5) samples were measured using a commercial multiplex bead array kit and an ELISA (for IFNα). Fold-change values are presented as the median ± IQR. Analyses were performed using a Kruskal-Wallis test with a Dunn’s post-hoc test. * *p* < 0.05.

**Figure 4 pathogens-12-00596-f004:**
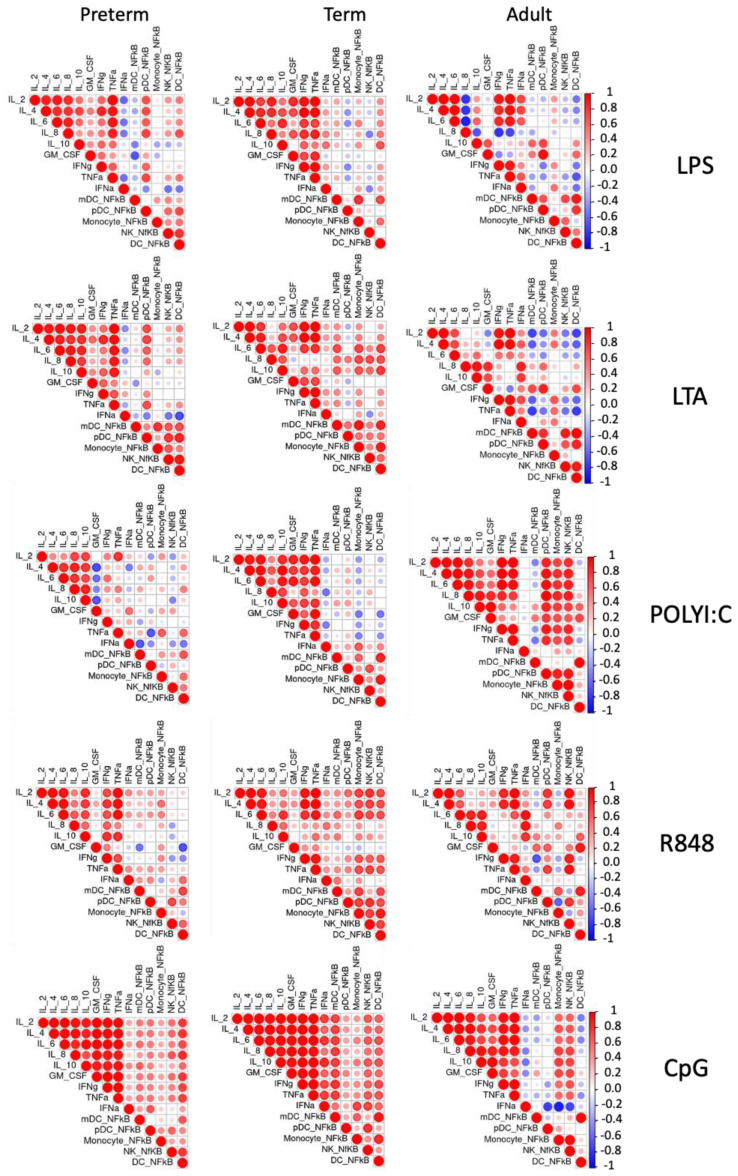
Correlation matrices following bacterial and viral TLR ligand stimulation. NF-κB MFI in immune cell subsets and cytokine production from preterm infants (n = 10), term infants (n = 10), and adults (n = 5) following LPS, LTA, poly I:C, R848, and CpG stimulation. Data is presented as correlation matrix plots, where red indicates positive correlation and blue indicates negative correlation between the variables. Large circles show a strong correlation, and small circles show a weak correlation between the variables. Correlation was determined by Pearson’s correlation.

**Table 1 pathogens-12-00596-t001:** Toll-like receptor (TLR) stimulation conditions.

TLR Type	TLR Ligand	Concentration	Manufacturer
TLR2	Lipoteichoic acid (LTA) from *Staphylococcus aureus*	2 µg/mL	Sigma Aldrich, St. Louis, MO, USA
TLR3	Polyinosinic-polycytidylic acid (poly I:C)	10 µg/mL	Invivogen, San Diego, CA, USA
TLR4	Lipopolysaccharide (LPS) from *Escherichia coli* O55:B5	10 ng/mL	Sigma Aldrich, St. Louis, MO, USA
TLR7/8	Resiquimod (R848)	2 µg/mL	Invivogen, San Diego, CA, USA
TLR9	CpG oligodeoxynucleotide 2216, class A (CpG-A ODN 2216)	2 µM	Invivogen, San Diego, CA, USA

**Table 2 pathogens-12-00596-t002:** Preterm infant and term infant characteristics.

	Preterm (N = 10)	Term (N = 10)	Adults (N = 5)
Gestational age (weeks) ^1^	33.4 (30.4–34.1)	38.8 (37–39.5)	-
Age	-	-	28–40 years
Sex ^2^	5F, 5M	1F, 9M	3F, 2M
Vaginal birth ^3^	10/10 (100)	10/10 (100)	-
Clinical chorioamnionitis	0/10 (0)	0/10 (0)	-

^1^ Data presented as median (range); ^2^ Female, F; Male, M; ^3^ Data presented as n (%).

## Data Availability

The data presented in this study are available on request from the corresponding author.

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
