# Peer review of "TLR Responses in Preterm and Term Infant Cord Blood Mononuclear Cells"

_pathogens, 2023, doi:10.3390/pathogens12040596_

Round 1

Reviewer 1 Report

Introduction

Line 43-45: Could the authors specify whether the diseases are more bacterial or viral?

Line 54: TLR activation is not always associated with protection. It could be responsible for tissue injury and contribute to disease.

Methods

Line 84: please add a brief description for PBMC separation and preservation method.

Please indicate whether you measured the PBMC purity and vitality before stimulation.

Line 105: is the volume of 900 mL correct? Do you mean µL?

Lines 107-108: why did the authors select different stimulation times (2 and 4 h) for bacterial and viral TLR ligands?

Please add a reference for selecting the TLR ligand concentration for the stimulation assay.

Line 117: is the volume of staining Ab (50mL) correct? The same for line 119.

Line 120: please add the composition of the permeabilisation buffer.

Results

TLR expression in preterm and term infants: please use “in” instead of “on” for TLR expression because some TLR are expressed intracellular.

Figure 1: are the cells here stimulated with bacterial TLR ligands? If not please correct the title of the figure.

Discussion

Did the authors compared the blood immune cell composition between preterm and term and adults? This could be included in the discussion and interpretation of the results.

What about the expression of TLR in blood lymphocytes like gd T cells and may these cells produce cytokines in response to TLR stimulation? The authors may include this point in the discussion of the correlation between NFkB activation and cytokine response to viral ligands.

Would the authors suggest doing the stimulation on purified cell populations instead of the whole PMBC?

Author Response

Thank you for your valuable time reviewing our manuscript. Please see the attachment for our responses. 

Reviewer 2 Report

The authors explore Toll-like receptor responses as a potential cause of increased susceptibility to bacterial and viral diseases in pre-term infants.  The authors collected cord blood samples from healthy pre-term infants for comparison with cord blood from healthy term infants and peripheral blood from healthy adults.  Using flow cytometry, they measured the surface expression of TLR proteins in immune cell subsets and characterized NF-kappa B activation following stimulation with TLR ligands.  They also determined cytokine levels following stimulation with TLR ligands using a multiplex bead array kit or ELISA.  They also presented correlation matrices between cytokine responses and NF-kappa B expression. 

For this reviewer, the main criticism of the manuscript relates to the statistical approaches. 

1.       Line 138-147.  The authors chose a non-parametric Kruskal-Wallis test over a parametric ANOVA approach.  They appear to make pairwise comparisons but do so without describing a post-hoc test like Dunn’s post-test.  Kruskal-Wallis tests, by themselves, cannot be used for pairwise comparisons in this situation where there are three or more comparisons being made.  For paired comparisons, post-hoc tests like Dunn’s (or another appropriate test) need to be completed.  Additional statistical analyses are required to support the conclusions the authors reach. 

2.       In presenting the data for figure 2, the authors described subtracting unstimulated mean fluorescence intensity (MFI) values from stimulated MFI values and then presented the data as median +/- IQR.  It is confusing to understand how the subtraction of a mean from another mean yields a median value.  Further explanation or clarification is needed. 

3.       In figure 3, the authors share fold change data (y-axis) on the log scale.  Would it be appropriate to transform these data to the log values for statistical analysis?  The authors should consult a statistics expert to identify the best approach. 

It may benefit the manuscript if the authors could address the following points.

4.       Lines 90-94.  Can more information be provided about the 5 healthy adults from whom the peripheral blood was collected?  Was the blood collected in the same region as the cord blood?

5.       Are the volumes listed on lines 105, 117, 121, and 124 (and perhaps other places in the manuscript) typos?  For example, it does not seem correct that 1 million cells were diluted in “900mL of R10 media.”

6.       Lines 108 and 109.  Why were these times chosen?  Can you provide justification for these timepoints?

7.       Figure 1.  In measuring the baseline TLR expression, did the authors collect any data regarding endosomal expression of TLRs?  Can the authors address the issue of surface vs. endosomal expression and if it may affect the study conclusions? 

Author Response

(The authors gave the same response as above.)

Round 2

Reviewer 2 Report

The authors have submitted an improved revision that addresses many of the concerns raised in the original review.  Some minor issues with grammar exist.  

Author Response

Thank you for the comment. We have made numerous changes to the grammar to fix the minor issues.